# Isolation and Characterisation of Quercitrin as a Potent Anti-Sickle Cell Anaemia Agent from *Alchornea cordifolia*

**DOI:** 10.3390/jcm11082177

**Published:** 2022-04-13

**Authors:** Olayemi Adeniyi, Rafael Baptista, Sumana Bhowmick, Alan Cookson, Robert J. Nash, Ana Winters, Jianying Shen, Luis A. J. Mur

**Affiliations:** 1Institute of Biological, Environmental and Rural Sciences, Aberystwyth University, Aberystwyth SY23 3DA, UK; ola8@aber.ac.uk (O.A.); rafaelb4@gmail.com (R.B.); sub23@aber.ac.uk (S.B.); akc@aber.ac.uk (A.C.); alg@aber.ac.uk (A.W.); 2Biochemistry Unit, Department of Science Technology, The Federal Polytechnic, Ado-Ekiti 360231, Nigeria; 3PhytoQuest Ltd., Plas Gogerddan, Aberystwyth SY23 3EB, UK; robert.nash@phytoquest.co.uk; 4Artemisinin Research Center, Institute of Chinese Materia Medica, China Academy of Chinese Medical Sciences, Beijing 100700, China

**Keywords:** sickle cell anaemia, *Alchornea cordifolia*, quercitrin, sickling metabolomics

## Abstract

*Alchornea cordifolia* Müll. Arg. (commonly known as Christmas Bush) has been used traditionally in Africa to treat sickle cell anaemia (a recessive disease, arising from the S haemoglobin (Hb) allele), but the active compounds are yet to be identified. Herein, we describe the use of sequential fractionation coupled with in vitro anti-sickling assays to purify the active component. Sickling was induced in HbSS genotype blood samples using sodium metabisulphite (Na_2_S_2_O_5_) or through incubation in 100% N_2_. Methanol extracts of *A. cordifolia* leaves and its sub-fractions showed >70% suppression of HbSS erythrocyte sickling. The purified compound demonstrated a 87.2 ± 2.39% significant anti-sickling activity and 93.1 ± 2.69% erythrocyte sickling-inhibition at 0.4 mg/mL. Nuclear magnetic resonance (NMR) spectra and high-resolution mass spectroscopy identified it as quercitrin (quercetin 3-rhamnoside). Purified quercitrin also inhibited the polymerisation of isolated HbS and stabilized sickle erythrocytes membranes. Metabolomic comparisons of blood samples using flow-infusion electrospray-high resolution mass spectrometry indicated that quercitrin could convert HbSS erythrocyte metabolomes to be like HbAA. Sickling was associated with changes in antioxidants, anaerobic bioenergy, and arachidonic acid metabolism, all of which were reversed by quercitrin. The findings described could inform efforts directed to the development of an anti-sickling drug or quality control assessments of *A. cordifolia* preparations.

## 1. Introduction

Sickle cell anaemia (SCA) is an autosomal recessive genetic blood disorder arising from the S allele of haemoglobin (Hb). SCA is prevalent in the tropics and especially in sub-Saharan Africa, where the S allele may confer some tolerance to malaria. The World Health Organization (WHO) estimates that 300,000 children are born with SCA annually, 75% of whom are in sub-Saharan Africa [1].

SCA arises from a single amino acid substitution of glutamic acid with hydrophobic valine in the Hb β-globin chain. This results in an altered haemoglobin tetramer (α_2_β_2_), haemoglobin S (HbS) [2]. HbS polymerises under hypoxic conditions, due to hydrophobic interactions between β6Val on two deoxy-HbS molecules [3]. The polymer of helical fibers lengthen and stiffen to cause the characteristic sickle shape of HbSS erythrocytes [4]. Polymerisation is linked to a dysregulation of cation homeostasis resulting from the activation of some ion channels, particularly the K^+^/Cl^−^ co-transport system and the Ca^2+^ dependent K^+^ channel (Gardos channel). Ca^2+^ activation of the Gardos channel increases H_2_O and K^+^ efflux, leading to the dehydration of sickled erythrocytes [5]. Haemoglobin is denatured to form hemichromes, histidine-linked complexes, on the internal surface of the membrane. The haem group releases Fe^3+^ to foster an oxidizing microenvironment [6].

The effects of SCA can be mitigated through episodic blood transfusions to stabilize the Hb levels [7], increasing the provision of oxygen [8], and through rehydration with intravenous fluids [9]. The pain associated with SCA crises can be managed with nonsteroidal anti-inflammatory drugs (NSAIDs) or other non-opioid analgesics [10]. However, there are relatively few chemical agents that interfere with the mechanism and/or kinetics of the sickling process, with hydroxyurea and voxelotor being most often used [11,12,13]. Such therapies have attendant limitations [14,15,16], especially their high cost and low availability for millions of patients in sub-Saharan Africa [17], as well as attendant risks with long-term clinical use [17,18,19]. Therefore, there is a need for new cost-effective anti-sickling small molecules to treat SCA.

Medicinal plants are widely used in Sub-Saharan Africa to manage SCA and have driven research into their active components. Thus, phenylalanine and *p*-hydroxy benzoic acid (PHBA) from *Cajanus cajan* [20]; zanthoxylol [21], betulinic acid [22], divanilloylquinic acids from *Fagara zanthoxyloides* Lam. (Rutaceae) [23]; butyl stearate from *Ocimum basilicum* [24]; and ursolic acid from *Ocimum gratissimum* L. (Lamiaceae) [25] have been linked to reduced sickling. Leaves of *Alchornea cordifolia* have been used as a “blood tonic” to reduce the symptoms of SCA in Nigeria. It has featured in some research that have characterised its biochemistry [26,27,28,29,30], but the active compounds linked to anti-sickling activities have not been identified.

In this paper, we described the isolation and characterisation of quercitrin from the Nigerian shrub, *A. cordifolia**,* as the main anti-sickling agent. Quercitrin was able to prevent and reverse in vitro HbSS erythrocyte sickling, primarily through the inhibition of HbS polymerization and membrane stabilization under hypoxic conditions.

## 2. Materials and Methods

### 2.1. Chemicals

LC-MS-grade formic acid, water, and acetonitrile and HPLC-grade solvents (methanol, dichloromethane, acetonitrile, ethyl acetate, and *n-*hexane and nitrogen (N_2_)) were all obtained from Fisher Scientific (Leicestershire, UK). Sodium metabisulfite (Na_2_S_2_O_5_), rutin, gallic acid, linoleic acid, 1,1-diphenyl-2-picrylhydrazyl (DPPH), ascorbic acid, quercetin, 2,2′-azobis(2-amidinopropane) dihydrochloride (AAPH), phosphate-buffered saline (PBS) pH 7.4, *p*-hydroxybenzoic acid (PHBA), L-phenylalanine, sodium chloride (NaCl), trifluoroacetic acid (TFA), Triton X-100 sodium dihydrogen phosphate, sodium chloride and L-phenylalanine were all purchased from Sigma-Aldrich (Gillingham, UK).

### 2.2. Collection of Plant Samples

*A. cordifolia* leaves were harvested from a bush at Ado-Ekiti, South-West, Nigeria, in May 2015 (Appendix A). Samples were deposited as a voucher specimen (UHAE2020030) at the Herbarium, Department of Botany, Ekiti State University, Nigeria. The leaves were shade-dried for 3 weeks until completely dehydrated.

### 2.3. Blood Sampling

HbSS blood samples from a single clinically diagnosed SCA sufferer (the first author (female, 32 years old), subject to informed consent) was used to evaluate the anti-sickling activities of the plant extracts. Samples of 5 mL of blood were extracted using a lavender topped vacutainer (BD Vacutainer tubes, ISS Ltd., Bradford, UK), which uses dipotassium/tripotassium salts of EDTA as an anticoagulant. Control HbAA samples were taken from a single volunteer (the corresponding author (male, 55 years old), subject to informed consent). Separate blood samples were taken for each experiment presented. The whole blood (4 mL) was centrifuged at 2800× *g* for 10 min at 4 °C to sediment the erythrocytes. The plasma supernatant was removed, and erythrocytes underwent sequential centrifugations (2800× *g* for 10 min at 4 °C) involving washing three times in phosphate buffered saline (PBS) at pH 7.4. The cells were finally resuspended in 5 volumes PBS to 1 packed volume of erythrocytes and were used immediately.

### 2.4. Extraction of A. cordifolia Leaves

Pulverized air-dried leaves of *A. cordifolia* (1 kg) were extracted 12.5 L dichloromethane (CH_2_Cl_2_; DCM), and then sequentially with 51.5 L of 75% methanol (75% MeOH; ALM (*Alchornea* methanol extract)), and two rounds of 10 L of sterile deionised water (H_2_O; “Aqueous”, “Aqueous2”) at room temperature with constant stirring. Each extraction occurred for a 72-h period. After filtration, the extracts were concentrated under reduced pressure at 40 °C and were stored at −20 °C until further use. The aqueous extract was freeze-dried.

### 2.5. Bioactivity-Guided Purification of MeOH Extract

Partial purification of MeOH extract (ALM; 96.5 g) involved using a modified method, as described by [31] (Appendix A). The crude extract of *A. cordifolia* was separated using silica gel (243.43 g adsorbent, 70–200 mesh, Material Harvest, UK) packed in a 40 × 500 mm (width x length) chromatographic column and eluted with a continuous solvent gradient of increasing polarity—*n-*hexane-ethyl acetate (EtOAc) (0–100%) and then EtOAc-methanol (0–100%). Reflecting differences in composition (as indicated by thin layer chromatography), 19 fractions were obtained (ALM1–ALM19). Thin layer chromatography (TLC) was performed on Sigma-Aldrich silica gel 60 F254 gel plates, and were visualized under UV light and by spraying sulphuric acid/MeOH (1:1) followed by heating.

Fractions 7 (ALM7) and 8 (ALM8) were combined and separated to yield 23 sub-fractions (ALM7A–ALM7W) on a silica gel-packed chromatographic column (70–200 mesh), and were eluted with a solvent gradient of increasing polarity—*n-*hexane-EtOAc (0–100%) and then EtOAc-methanol (0–100%).

Following the anti-sickling assays, ALM7T was further separated by preparative HPLC using a C18 3.5 m, 4.6 × 50 mm column (Waters, Borehamwood, UK). The eluting gradient was as follows: 90% water and 10% acetonitrile for 2.5 min, then 100% acetonitrile at 8.5 min and continuing at 100% until 13 min. There was 0.01% trifluoroacetic acid present throughout and the flow rate was 1.5 mL/min. This yielded eight peaks, separated into single fractions, ALM7T1–ALM7T8.

### 2.6. Ultra High-Performance Liquid Chromatography–High Resolution Mass Spectrometry (UHPLC-HRMS)

The fractions were analysed on an Exactive Orbitrap (Thermo Fisher Scientific, Waltham, MA, USA) mass spectrometer, which was coupled to an Accela Ultra High-Performance Liquid Chromatography (UHPLC) system (Thermo Fisher Scientific). Chromatographic separation was performed on a reverse phase (RP) Hypersil Gold C18 1.9 µm, 2.1 × 150 mm column (Thermo Scientific) using H_2_O using 0.1% formic acid (*v*/*v*, pH 2.74) as the mobile phase solvent A and ACN/isopropanol (10:90) with 10 mM ammonium acetate as mobile phase solvent B. Each sample (10 μL) was analysed using a 0–20% gradient of B from 0.5 to 1.5 min, and then to 100% in 10.5 min. After 3 min of being isocratic at 100% B, the column was re-equilibrating with 100% A for 7 min.

### 2.7. Nuclear Magnetic Resonance

NMR spectra were obtained using a Bruker Ultra shield-500 NMR spectrophotometer (^1^H-NMR 500 MHz, ^13^C-NMR 100 MHz) using MeOD as the solvent reference.

Quercitrin [2-(3,4-dihydroxyphenyl)-5,7-dihydroxy-3-(((2S,3R,4R,5R,6S)-3,4,5-trihydroxy-6-methyltetrahydro-2H-pyran-2-yl)oxy)-4H-chromen-4-one] (Figure 1): Yellow powder; *m/z* 447.09363 [M − H]^+^ (calcd. For C_21_H_20_O_11_, 448.100561). ^1^H-NMR (500 MHz, MeOD): δ 0.95 (^3^H, d, J = 6.0 Hz), 3.32 (^1^H, m), 3.43 (^1^H, m), 3.76 (^1^H, dd, J = 3.0 and 3.0 Hz), 4.23 (^1^H, s), 5.36 (^1^H, s), 6.20 (^1^H, d, J = 1.8 Hz), 6.37 (^1^H, d, J = 1.8 Hz), 6.91 (^1^H, d, J = 8.4 Hz), 7.30 (^1^H, dd, J = 8.4 and 2.1 Hz), and 7.34 (^1^H, d, J = 2.0 Hz) ppm. ^13^C-NMR (100 MHz, MeOD): δ 17.67, 71.96, 72.05, 72.24, 73.38, 94.79, 99.88, 103.58, 106.00, 116.45, 117.07, 122.95, 123.10, 136.31, 146.42, 149.79, 158.56, 159.35, 163.22, 165.83, and 179.70 ppm. These spectroscopic data agreed with previous studies for quercitrin [32,33].

### 2.8. In Vitro Sickling-Inhibition and Reversibility Assays

The sickling-inhibition assay consisted of 100 μL of HbSS erythrocytes, 100 μL of PBS, and 100 μL of the test extract, and was incubated at 37 °C for 2 h. To induce sickling, freshly prepared 2% (*w*/*v*) Na_2_S_2_O_5_ solution (300 μL) was incubated with the cells for an additional hour in a water bath at 37 °C. The cells were then fixed with 3 mL of 5% (*w*/*v*) buffered formalin solution. A total of 10 µL of the incubated cells were transferred to a haemocytometer and five fields were counted on each slide using a Leica ATC 2000 Binocular Phase Contrast Microscope at 40× magnification. The cells were classified as either normal or sickled. Each assay was repeated five times to generate the data presented.

For the reversibility assays, the cells were prepared as above and were incubated with 2% (*w*/*v*) Na_2_S_2_O_5_ at 37 °C for 1 h. Then, 100 μL of each sample was added and incubated at 37 °C for an additional 4 h period. The cells were fixed and mounted on a haemocytometer and were counted as described earlier.

### 2.9. Erythrocyte Leakage Assay

Sterile microcentrifuge tubes were each filled with 1 mL HbSS erythrocytes and centrifuged at 2800× *g* for 10 min at 4 °C. The supernatant was discarded, and the erythrocytes were washed three times with phosphate buffered saline (PBS; 0.01 M pH 7.4) and resuspended in 4% *v/v* PBS. Samples of 90 μL were added to the wells on a 96-well plate. Then, 10 µL of the fraction or chemical composition tested (at 10× final concentration) was added to wells of the first row. These were then serially (1 in 10) diluted down to row 6. Row 7 contained erythrocyte suspensions with 0.1% Triton-X 100 (Sigma-Aldrich UK) as a positive control and row 8 contained only erythrocytes as the negative control. The plate was incubated for 1 h at 37 °C. Following a 5 min centrifugation at 2800× *g* at 4 °C, 70 µL of supernatant from each well was transferred to a transparent, flat bottom 96-well plate. Changes in absorbance (OD 450 nm), indicating haemoglobin leakage, were measured using the Hidex Sense Plate Reader (LabLogic, Sheffield, UK). This absorbance was used to calculate the percentage haemolysis (0.1% triton X-100 = 100% and the negative control = 0%). Experiments were performed in quadruplicate.

### 2.10. Hb Polymerisation Assay

Hb polymerisation assays [34] were adapted for 96-well plates to assess polymerized Hb SS turbidity. A haemolysate was prepared by adding 2 mL of ice-cold distilled water to packed, washed erythrocytes, and then the cellular debris was pelleted by centrifugation at 6000× *g* for 20 min at 4 °C. Then, 220 µL of 2% Na_2_S_2_O_5_, 20 µL of the test compound (at five different concentrations and using PBS as the control), and 50 µL HbSS haemolysate (1:5 *v/v* dilution in PBS) were added into a 96-well plate and incubated. The 96-well plate was shaken and the absorbance at 700 nm was taken in 30 s intervals for period of 20 min (Hidex sense microplate reader, LabLogic, UK). The tests were carried out in quadruplicate.

### 2.11. Scanning Electron Microscopy (SEM)

Erythrocytes were fixed in 2% glutaraldehyde PBS for 30 min, and were rinsed three times in a 0.075 M sodium phosphate buffer (pH 7.4). The samples were then incubated with 2% OsO_4_ in PBS (pH 7.4) for 2 h at 4 °C and then rinsed thrice in a 0.075 M sodium phosphate buffer (pH 7.4). Subsequently, the sample was dehydrated in 30%, 50%, 70%, 90%, and finally, underwent three changes of 100% ethanol. For SEM, 200 μL of the samples in hexamethyldisilazane (HMDS) were air dried on coverslips, coated with carbon, and imaged using a Zeiss Ultra plus FEG SEM.

### 2.12. Metabolomic Analyses

Samples of 500 μL washed HbSS erythrocytes, 500 μL of PBS, and 500 μL of quercitrin (0.5 mg/mL in final volume) were mixed and incubated at 37 °C for 2 h. Sickling was induced through chronic deoxygenation, with the reaction mixture (1500 μL) in an anaerobic tube deoxygenated by gently bubbling 100% N_2_ through in a 37 °C water bath for 12 h. After the incubation period, the morphology of the cells was confirmed using light microscopy.

Erythrocyte extractions were carried out using published protocols [35,36]. Extracts were transferred to a 2 mL microcentrifuge tube, and were dried using a SpeedVac at 4 °C. The pellets were then resuspended in 100 µL of 50% methanol, in a HPLC vial containing a 0.2 mL flat bottom micro insert for flow infusion electrospray ion high resolution mass spectrometry (FIE-HRMS) analysis. FIE-HRMS was performed with an Exactive HCD mass analyser equipped with an Accela UHPLC system (Thermo-Scientific, UK). Data acquisition for each individual sample was conducted in alternating positive and negative ionisation mode, over four scan ranges (15–110, 100–220, 210–510, and 500–1200 *m/z*) with an acquisition time of 5 min. Individual metabolite *m/z* values were normalised as a percentage of the total ion count for each sample. The derived data are provided in Appendix A. Data were normalised to total ion count and log_10_-transformed. Metabolites and pathway identification were performed by the MetaboAnalyst 4.0 MS peaks to pathway algorithm [37] (tolerance = 5 ppm, reference library: Homo sapiens). This involved metabolites being annotated using the KEGG database, considering the following possible adducts: [M]^+^, [M + H]^+^, [M + NH_4_]^+^, [M + Na]^+^, [M + K]^+^, [M − NH_3_ + H]^+^, [M − CO_2_ + H]^+^, [M − H_2_O + H]^+^; [M]**^−^**, [M − H]^−^, [M + Na − 2H]^−^, [M + Cl]^−^, and [M + K − 2H]^−^. For each *m/z,* the annotation was made using a 5 ppm tolerance on their accurate mass and considering the different adducts formed for each metabolite.

### 2.13. Statistical Analysis

The statistical analyses used SPSS version 26.0 software and XLSTAT_2020.1.1.64525. One way analysis of variance (ANOVA) coupled with Tukey’s post-hoc test were used to compare the data and to identify means with significant differences; *p* values of <0.05 were considered significant.

## 3. Results

### 3.1. Sample Extraction and Anti-Sickling Activity of A. cordifolia Crude Extracts

Sequential extractions using DCM, 75% MeOH, and 100% water from the dried leaves derived samples of 2.47, 9.65, and 8.28% dry weight. The sickling-inhibitory activities of *A. cordifolia* leaf extracts were compared at 1 mg/mL in PBS on a haemocytometer (Appendix A and Figure 2A,B). The MeOH extract exhibited a significantly (*p* < 0.01) higher sickling-inhibitory activity (91.4%) than any other extract (Figure 2A,B). MeOH extract also exhibited a better sickling reversibility activity than any other extract (Figure 2C). The cytotoxicity of aqueous and MeOH leaf extracts of *A. cordifolia* on HbSS erythrocytes was evaluated using an erythrocyte leakage assay (Appendix A). Some haemolysis was observed in both the MeOH and aqueous extracts at 10 mg/mL, but this was only < 2% at 1 mg/mL.

### 3.2. Isolation of the Anti-Sickling Bioactives in the A. cordifolia Methanolic Extract

A bioassay-guided purification process was used to isolate the bioactives in the MeOH extract of *A. cordifolia* (ALM; 50 g) (Appendix A). Of a total of 19 fractions, three fractions, ALM5 (0.2% yield), ALM7 (0.6% yield) and ALM8 (5.3% yield), exhibited sickling inhibition above 95% at 1 mg/mL concentration, especially ALM7 (96.5 ± 2.8%) and ALM8 (96.7 ± 2.7%). ALM7 and ALM8 were combined due to their chemical similarities, as revealed by TLC. A total of 23 sub-fractions (ALM7A–ALM7W) were fractionated from combined ALM7 and ALM8. As enrichment of the bioactive product was expected following fractionation, the sickling inhibiting activities were assessed at a lower concentration range than previously (Appendix A). Sickling-inhibiting activities of >70% were observed with 0.25 mg/mL in sub-fractions ALM7N, ALM7Q, ALM7T, and ALM7V.

Sub-fraction ALM7T was further separated by preparative-HPLC into individual compounds to yield ALM7T1–ALM7T8. Screening for anti-HbSS erythrocyte sickling properties indicated that ALM7T5 showed the best sickling-inhibition activity (Figure 3). Representative haemocytometer images can be seen in Appendix A. NMR was used to identify the only chemical detected within these peaks. The chemical in ALM7T5 was identified as quercitrin (quercetin-3-rhamnoside), a flavonol glycoside, based on high resolution LC-MS and NMR data.

The ability of quercitrin to reverse sickling in erythrocytes in Na_2_S_2_O_5_-induced hypoxia after a 4 h incubation period was tested. The highest concentration tested, 0.80 mg/mL, could reverse sickling (41.8% ± 4.8%) and activity was still detected (18.1% ± 1.2%) at 0.05 mg/mL (Figure 4).

The biological activity of quercitrin has been previously associated with its aglycone, quercetin [38,39]. Therefore, the anti-sickling properties of quercetin and quercitrin were compared across a concentration range (Appendix A). The results showed a significantly (*p* < 0.01) higher anti-sickling activity (ranging between 93.1% ± 1.6% to 36.9% ± 1.9%) in quercitrin compared to quercetin (ranging between 11.8% ± 0.98% to −1.14% ± 1.0%). This suggested that quercitrin, but not quercetin, was able to decrease erythrocyte sickling.

Different concentrations of quercitrin (0.25–4 mg/mL) were tested for their ability to inhibit the polymerisation of deoxygenated HbS over 20 min (Figure 5). Deoxy HbSS and HbAA haemolysates (with Na_2_S_2_O_5_ and phosphate buffers) were used as the controls. Quercitrin prevented HbS polymerisation over all of the tested concentrations so that there was no significant difference in HbAA results (Figure 5A). A comparison was also made with PHBA, known to inhibit sickling in erythrocytes [20,40] (Figure 5B). PHBA showed a similar ability to suppress the exhibited HbSS polymerization, but at 0.5 and 0.25 mg/mL, suppression was significantly (*p* > 0.05) less effective than quercitrin (Figure 5B). Quercetin exhibited no ability to prevent the polymerisation of HbSS, with no significant difference to the positive control (Appendix A). Taken together, these data indicate that quercitrin is a potent inhibitor of in vitro HbS polymerisation and this is most likely to represent its main mode of action in inhibiting and reversing HbSS-erythrocyte sickling.

### 3.3. Assessing the Impact on Quercitrin on HbSS Sickling Using Metabolomics

In using metabolomics to assess quercitrin’s mode of action, it was not possible to use 2% Na_2_S_2_O_3_ to induce the sickling phenotype, as this would dominate the subsequent metabolite profile. Thus, an alternative approach was employed where anoxic conditions were induced through the replacement of air with 100% N_2_. This was proven to be a highly successful approach, as demonstrated using SEM (Appendix A). Imposition of N_2_-induced anoxia led to sickling in HbSS erythrocytes, but not HbAA erythrocytes (data not shown). HbSS erythrocytes treated with 0.5 mg/mL quercitrin exhibited morphologies that were similar but not identical to the HbAA erythrocytes. Interestingly, erythrocytes treated with 0.5 mg/mL PHBA did all exhibit the normal discoid phenotype, with some sickled cells observed.

In the metabolomic treatments, HbSS erythrocytes were either (1) maintained under normoxic conditions (SS), (2) deoxygenated with N_2_ (SS-N) only, or (3) incubated with quercitrin followed by deoxygenation with N_2_ (SS-Q). Controls consisted of HbAA erythrocytes (AA) with no quercitrin treatment (Figure 6). Principal component analysis (PCA) indicated that the different experimental classes formed two clusters; one broadly associated with sickled cells the other with non-sickled cells (Figure 6A). SS and SS-N samples were both closely clustered across principal component 1 (PC1), which describes the major source of variation. This suggested pre-existing metabolomic changes in the HbSS erythocytes, even under normoxia. However, by adding quercitrin, the hypoxic HbSS metabolomes shifted so that the samples clustered with the HbAA group.

The sources of variation between sickled and non-sickled erythrocyte samples were determined (Figure 6B). We annotated metabolites using the KEGG database and their relative abundances were discriminated between the sickled and non-sickled phenotypes, as shown using a heatmap. The sickled group showed relative increases in metabolites such as arachidonic, stearic, myristic, and linolenic acids, which were suggestive of lipid processing. The sickled cells also appeared to be relatively deficient in glucose and fructose. These effects were all reversed by quercitrin treatment. To provide functional information for these differences, biochemical enrichment analyses were conducted. The over-representation analysis (ORA) method was used to evaluate pathway-level enrichment based on significant features (*p*-value is measured with Fisher’s exact test) [41], combined with gene set enrichment analysis (GSEA), which extracts biological meaning from a ranked metabolite list [42]. A total of six metabolic pathways were identified to be significantly enriched (Appendix A).

HbAA (AA) and HbSS (SS) erythrocytes were maintained under normoxic conditions or under N_2_ (N), and, in some cases, treated with quercitrin (0.5 mg/mL) (SS-Q) imposition of hypoxic conditions. (A) PCA of the derived metabolite profiles for each treatment (note the separation of the samples into two main clusters, reflecting sickled and non-sickled groups) (B) heatmap based on significant metabolite differences between the non-sickled (cluster 1) and sickled (cluster 2) groups

These suggested that the metabolomic switches in the erythrocyte between the sickled and non-sickled states apparently involves redox changes (ascorbate), thiol metabolism (cysteine), fatty acid processing, and haem metabolism (porphyrin).

## 4. Discussion

The high prevalence of SCA in the developing nations of West Africa is driving the requirement for cost-effective means to treat the disease. In Sub-Saharan Africa, medicinal plants are used widely to manage SCA, although relatively few have been validated scientifically. In the case of *A. cordifolia*, an aqueous extract showed an anti-sickling activity [43,44], but the bioactives had not been previously defined. In this study, we followed a bioactivity guided fractionation and purification strategy to define quercitrin as the main active anti-sickling agent in *A. cordifolia*.

### 4.1. Inhibition of HbS Polymerisation: An Important Mode of Action for Quercitrin

Beyond simply defining quercitrin as the anti-sickling chemical, we considered its mode of action. Quercitrin has been previously demonstrated to have a low toxicity profile on mouse macrophages (IC50 of approximately 0.1 mg/mL) [45] and to have potent antioxidant [46,47], antiapoptotic [48,49], anti-leishmanial [50], anti-diarrhoeal [51], anti-nociceptive [52], and anti-inflammatory activities [53].

Many anti-sickling drug leads effect the Hb gene [54] or the HbS molecule, or are erythrocyte membrane modifiers [55]. Our methods did not assess Hb gene modification, but we have provided clear evidence that quercitrin affected the HbS protein to suppress polymerisation. This was important, as clinically, the delay of HbS polymerization during the transit of erythrocytes through post-capillary venules is necessary for SCA disease remediation [56]. Our assessments of extracted Hb polymerisation showed the expected results for the controls, i.e., no polymerisation with HbA (HbAA), oxygenated HbS (HbSS), but deoxyHbS did polymerise. However, with quercitrin, deoxyHb polymerisation was suppressed in a concentration-dependent manner, comparable to PHBA. Crucially, a recent study used multi-spectroscopic and molecular simulation techniques and showed that isoquercitrin had an anti-sickling activity through a direct interaction with haemoglobin molecules [57]. Thus, quercitrin could act by stereospecific covalent or non-covalent attachment to the HbS molecule [58]. Additionally, quercitrin was able to partially reverse erythrocytes sickling after a two-hour incubation period, suggesting quercitrin was able to modify already polymerized HbS and thus have some ability to reverse polymerisation. This partial effect may reflect the rate at which quercitrin is transported across the membrane.

The pharmacological activities of a given bioactive compound are dependent on its chemical structure [59]. Previous investigations have credited the biological activity of quercitrin to the aglycone, quercetin [38,39], but the possibility of the glycosidic residue being crucial for its effects as the glycoside activity has also been suggested [39,60]. The quercetin aglycone or glucoside is not found in human plasma, however conjugates, such as quercetin-3-glucuronide, quercetin-3′-sulfate, and isorhamnetin-3-glucuronide, have been found [61]. It is generally thought that flavonoid glycosides such as quercitrin, enter the colon and are hydrolysed to the aglycone by quercitrinase found in Enterobacteria [62]. The aglycone is then absorbed in the large intestine easily because of its lipophilicity, and is then metabolized in the liver by O-methylation, glucuronidation, and/or sulfation [63]. We assessed if quercetin could be an active anti-sickling agent, as suggested by Muhammad et al. [64]. However, we failed to demonstrate any appreciable anti-sickling activities for quercetin, thus the substitution of an alpha-L-rhamnosyl moiety at position 3 via a glycosidic linkage in quercitrin would be important for its biological activity. Confirmation of this mode of action could focus on use of X-ray crystallography of Hb-quercitrin to demonstrate this.

### 4.2. Insights into the Mechanism of Quercitrin Effects Using Metabolomics Studies

To provide a wider appreciation of the effects of quercitrin, we applied an omic approach based on metabolite detection using direct infusion-high resolution mass spectroscopy. Multivariate statistical assessments of the derived data provided a series of important observations. Firstly, it was apparent that even under normoxia, the metabolomes of HbSS erythrocytes were unlike HbAA cells, and, indeed, were more like anoxic cells. This could suggest that even under normoxia, HbSS cells are poised to sickle and, indeed, the SEM of suggested cellular irregularities. However, the addition of quercitrin caused a shift in the HbSS metabolome, so that it became like HbAA. This allowed us to assess the metabolomic difference between sickled and non-sickled erythrocytes that quercitrin was effectively correcting.

The detection of porphyrin metabolites was predictable as the breakdown of haemoglobin that is a feature of sickling [6]. More mechanistic metabolite changes are suggested from the inositol phosphate metabolite, D-myo-inositol 1,4,5-trisphosphate (Ins (1,4,5) 3PO_4_). This inhibits human erythrocytes Ca^2+^-stimulatable, Mg^2+^-dependent adenosine triphosphatase (Ca^2+^-ATPase) activity [65], and the binding of calmodulin to the erythrocyte’s membrane [66]. Such effects could perturb the Ca^2+^ activated Gardos channel, leading to dehydration of HbSS-erythrocytes [6]. This suggestive data could be hinting at an additional role for quercitrin in influencing Ca^2+^ activated Gardos channels.

Another altered pathway involves bioenergetic metabolism. Glycerolipid pathways include monoacylglycerols (MAGs), diacylglycerols (DAGs), triacylglycerols (TAGs), phosphatidic acids (PAs), and lysophosphatidic acids (LPAs) with functions in energy generation [67]. The hexose monophosphate shunt, which is parallel to the glycolytic pathway, generates NADPH and pentoses, as well as ribose 5-phosphate, a nucleotides-synthesis precursor [68]. Increased D-glucose 1-phosphate, when interconverted to D-glucose 6-phosphate, will also feed through to the glycolytic pathway involved in the generation of ATP in the anabolic generation of intracellular energy [69]. This could indicate that the sickle cells could be exhibiting a tendency towards anaerobic respiration. Reduced glucose 1-phosphate levels in quercitrin-treated samples suggest that the anaerobic shifts are countered by quercitrin.

Sickle cells generate approximately two times more reactive oxygen species compared with normal red blood cells [70], and this is linked to endothelial dysfunction, inflammation, and multiple organ damage [71,72]. Decreased intravascular sickling has been linked with reduced oxidative stress and also increased nitric oxide bioavailability [73]. Such oxidative effects with sickling were suggested from the metabolomic analysis, as “ascorbate and aldarate” metabolism pathways were being affected in our experiments. It should be noted that glutathione, detected in our metabolome, also plays an important role in the anti-oxidative process. Glutathione protects the red cells from oxidative damage, denaturation of haemoglobin, the formation of Heinz bodies, reduced cell deformability, and intravascular haemolysis [74].

## 5. Conclusions

In this work, quercitrin was isolated from *A. cordifolia* and its anti-sickling activity was characterized. Quercitrin inhibits and marginally reverses HbSS erythrocytes under hypoxic conditions in vitro. Experimental evidence was presented to support its action against two features of SCA. Most importantly, quercitrin inhibited Hb polymerisation, but it also stabilized the HbSS-erythrocytes membrane to reduce erythrocytes’ fragility. Metabolomics was used to provide a wider description of the sickling process and the effects of quercitrin. All of these were reversed with quercitrin treatment.

## Figures and Tables

**Figure 1 jcm-11-02177-f001:**
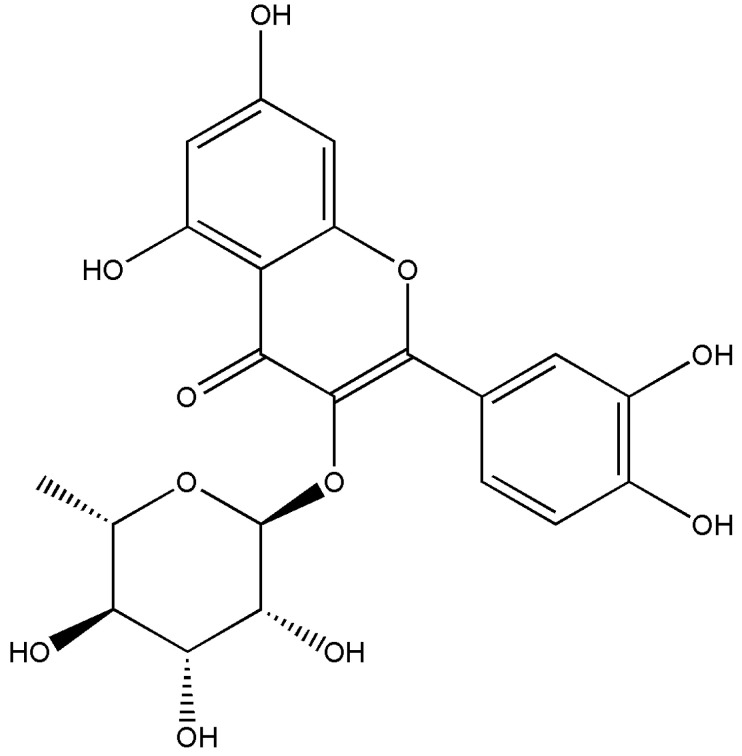
The chemical structure of quercitrin.

**Figure 2 jcm-11-02177-f002:**
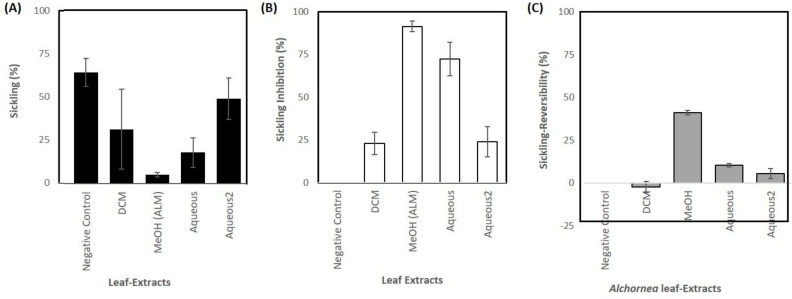
The effect of *Alchornea* spp. leaf extracts (1 mg/mL) on percentage sickling on incubation with erythrocytes in Na_2_S_2_O_5_-induced hypoxic conditions. (**A**) Percentage sickling, (**B**) percentage sickling inhibition, and (**C**) percentage reversion of sickling. The data represent the average of three similar results from the repeat experiments. The negative controls are the sickled erythrocytes in Na_2_S_2_O_5_-induced hypoxic conditions that were not treated with *Alchornea* spp. leaf extracts. DCM = dichloromethane extract. ALM = Alchornea methanol extract (75 % methanol: 25 % H_2_O).

**Figure 3 jcm-11-02177-f003:**
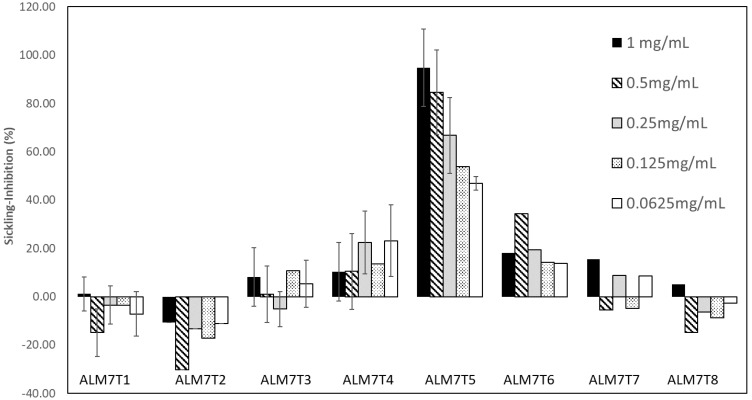
The effect of Alchornea methanol extract fractions of ALM7, designated ALM7T1-8 (see Appendix A) on ex vivo erythrocyte sickling in Na_2_S_2_O_5_-induced hypoxia. Treatments with all concentrations of ALM7T5 showed significant increases (*p* < 0.001) in sickling inhibition over zero.

**Figure 4 jcm-11-02177-f004:**
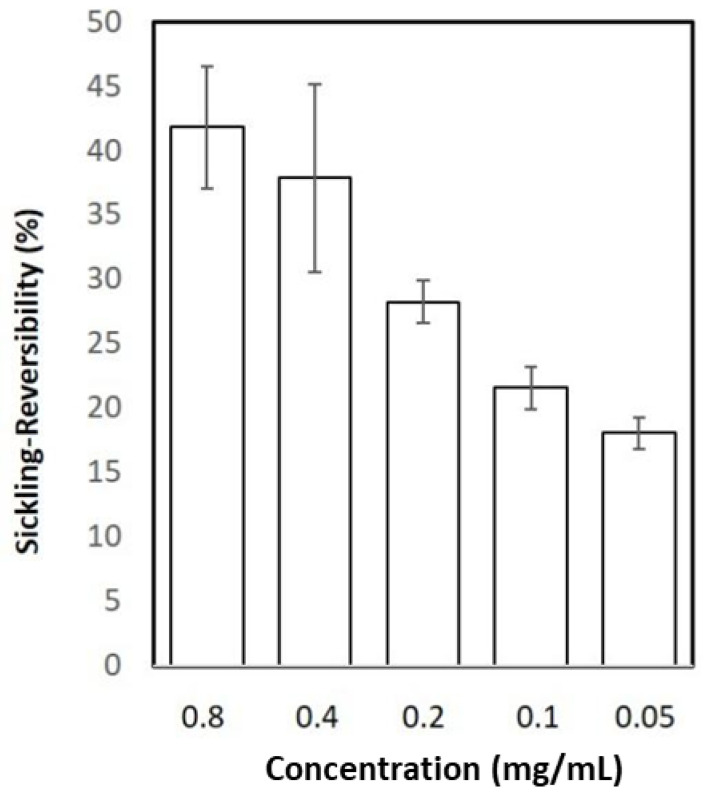
Reversibility effects of quercitrin on HbSS-RBC sickling in vitro, at low oxygen tension induced by Na_2_S_2_O_5_ after a 5 h-incubation period. The data are represented as mean and SD, obtained from three independent experiments. The treatments all showed significant increases (*p* < 0.001) in sickling reversibility over zero.

**Figure 5 jcm-11-02177-f005:**
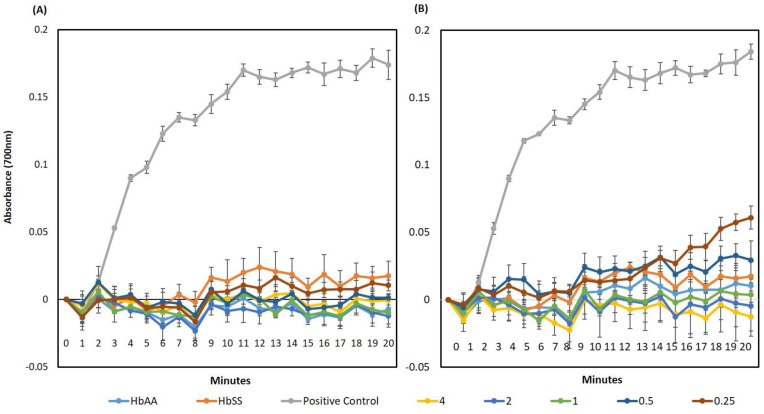
Inhibitory Effects of different concentrations of 0.25-4 mg/mL of (**A**) quercitrin and (**B**) p-hydroxybenzoic acid on HbS polymerisation, in vitro. Quercitrin inhibits HbS polymerisation compared to the “positive control”: deoxyHbS without quercitrin. With HbSS; HbS not subjected to deoxygenation. DeoxyHbA (HbAA) did not polymerise and represents a negative control. This represents data obtained from three typical independent experiments performed in quadruples.

**Figure 6 jcm-11-02177-f006:**
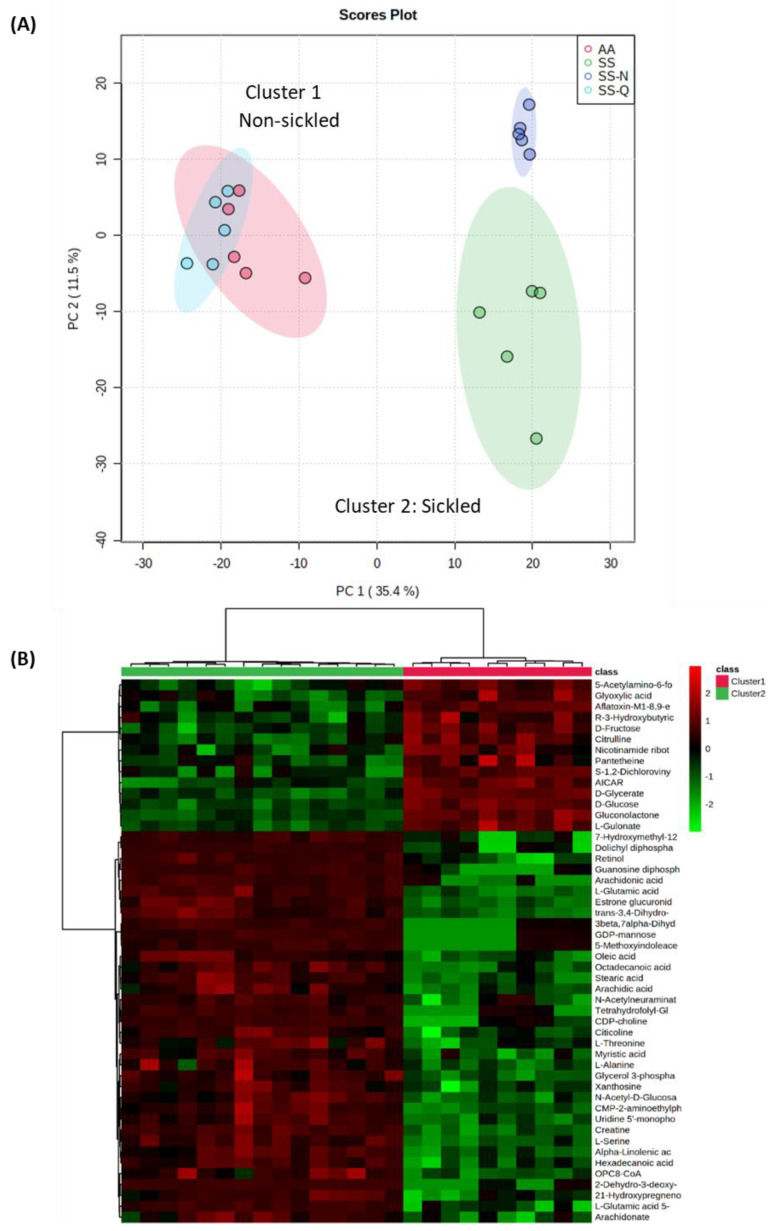
The metabolomic impact of quercitrin on erythrocyte sickling shown using (**A**) Principal component analysis (PCA and (**B**) hierarchical cluster analysis highlighting the major sources of variation between cluster 1 and cluster 2 on the PCA.

## Data Availability

All metabolomic data are supplied in the Appendix A.

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
