# Peer review of "Isolation and Characterisation of Quercitrin as a Potent Anti-Sickle Cell Anaemia Agent from Alchornea cordifolia"

_jcm, 2022, doi:10.3390/jcm11082177_

Round 1

Reviewer 1 Report

The manuscript was good in general. I have the following comments and suggestions that need to be collected or clarified.

  1. Is it possible to show the A. cordifolia tree figure in the texts or supplementary materials?
  2. Please include the blood sample characteristics such as gentle, age, number of blood samples and etc.
  3. Authors stated that “The sickling-inhibitory activities of A. cordifolia leaf-extracts were compared at different concentrations (1, 5 and 10 mg/mL in PBS)”. But Figures 2A, B, and C show only 1 mg/mL, How about 5 and 10 mg/mL?
  4. Figure 2A, B, and C, what is the difference between control and negative control?
  5. Figure 3 shows that there are %sickling inhibition is less than 0, please describe.
  6. Figures 3, 4, S3, S5, and S6 do not show control, please describe.
  7. Figure S4; a, b, c, d, e, and f on top of the bar, please describe.
  8. Please put the symbol on top of the significant bar in all figures.

Author Response

THANK YOU FOR YOUR CONSTRUCTIVE COMMENTS.  OUR RESPONSES ARE IN BLOCK LETTERS BELOW THE REVIEWERS COMMENTS.  MOST OF THE COMMENTS AROSE FROM THE FIRST AUTHOR NOT LONGER IN THE GROUP AND SOME ERRORS WERE INCLUDED IN THE TEXT. OUR SINCERE APOLOGIES.

The manuscript was good in general. I have the following comments and suggestions that need to be collected or clarified.

  1. Is it possible to show the A. cordifolia tree figure in the texts or supplementary materials?

YES, WE HAVE NOW SUPPLIED THIS INFORMATION IN SUPPLEMENTARY FIGURE 1.

Please include the blood sample characteristics such as gentle, age, number of blood samples and etc.

THIS INFORMATION HAS NOW BEEN ADDED TO THE METHODS. ACTUALLY MULTIPLE SAMPLES WERE USED FROM THE SAME VOLUNTEERS. THAT WAS NOT CLEAR IN THE ORIGINAL TEXT. OUR APOLOGIES FOR THIS ERROR

Authors stated that “The sickling-inhibitory activities of A. cordifolia leaf-extracts were compared at different concentrations (1, 5 and 10 mg/mL in PBS)”. But Figures 2A, B, and C show only 1 mg/mL, How about 5 and 10 mg/mL?

OUR APOLOGIES. WE HAVE REMOVED THE REFERALS TO 5 AND 10 MG/ML AS THEY SIMPLY SHOWED THE SAME TREND EXCEPT MORE CLEARLY. HOWEVER, WE FELL THAT THE DATA FOR 1 MG/ML IS THE BEST TO PRESENT AS IT SHOWS THE POTENCY OF THE RELATIVELY CRUDE EXTRACTS.

Figure 2A, B, and C, what is the difference between control and negative control?

THIS WAS INCLUDED AS AN ERROR. A THE “CONTROL” IS THE NEGATIVE CONTROL AND THE “CONTROL” COLUMN HAS BEEN DELETED.

Figure 3 shows that there are %sickling inhibition is less than 0, please describe.

WE HAVE REPLOTTED THESE DATA AS THEY WE INDEED POORLY PRESENTED. HOPEFULLY, IT IS NOW CLEARER THAT ALM7T5 SHOWED DOSE DEPENDENT SUPPRESSION OF SICKLING. IT IS THE CASE THAT SOME EXTRACTS APPEARED TO SLIGHT PROMOTE SICKLING, BUT GIVEN THE NARRATIVE OF THE PAPER, WE DECIDED THAT THIS WAS NOT IMPORTANT. WE ARE HAPPY TO CHANGE THIS IS THE REVIEWER INSISTS.

Figures 3, 4, S3, S5, and S6 do not show control, please describe.

IN EACH CASE THE NEGATIVE CONTROL VALUE IS “0” I.E. NO EFFECT ON SICKLING. WE DO NOT HAVE POSITIVE CONTROL DATA.

Figure S4; a, b, c, d, e, and f on top of the bar, please describe.

OUR APOLOGIES FOR OMITTING THIS. THE LETTERS INDICATED DATA SETS BETWEEN  WHICH THERE WAS NO SIGNIFICANT DIFFERENCE.

Please put the symbol on top of the significant bar in all figures.

OUR APOLOGIES FOR OMITTING THESE. THEY NOW HAVE BEEN ADDED, WERE WE COULD BUT IF THIS WAS THOUGHT TO “CLUTTER” THE FIGURE. THE RELEVANT DATA HAVE BEEN ADDED TO THE LEGENDS.

Reviewer 2 Report

It was my pleasure to review your manuscript.

Please correct the following:

p 62 & p 81  p-hydroxybenzoic  acid - italise p

p102 and subsequent citations. (75% MeOH; ALM) What is ALM please give full name of acronym.

p140 & 141 (1H-NMR 500 MHz, 13C-NMR 100 MHz) 1 & 13 should be superscripts  1H NMR  13C NMR.

p142 & 143 the systematic chemical name of Quercitrin is wrong. 

The IUPAC name is: 2-(3,4-dihydroxyphenyl)-5,7-dihydroxy-3-(((2S,3R,4R,5R,6S)-3,4,5-trihydroxy-6-methyltetrahydro-2H-pyran-2-yl)oxy)-4H-chromen-4-one

p143 447.09363 [M+H]+ this value indicates that the measured ion was [M-H] the + sign should superscript [M-H]+

p144 C21H20O11 the numbers in molecular formulae should subscripted C21H20O11

p144 change to 1H NMR & p147 to 13C NMR

p145  3.76 (1, dd, J=3.0 and 3.0 Hz) should be 3.76 (1H ... also the coupling constants of a double doublet cannot be of the same value. J values are different (big and small). It would be good to provide both 1H and 13C NMR spectra as pdfs in the Supplementary Section to verify purity of the compound. 

p170  change to Samples of 90 µL were added

p171 chemical being change to chemical composition

p192 change to OsO4

p218 & 219 the numbers in chemical formulae should be subscripted and the + and - signs of the molecular ions should superscripted.

p247 Na2S2O5- change to Na2S2O5

p318 PCA give full name of the acronym here.

p370-373  This sentence does not read right and should be rephrased "It was important as clinically, the delay of HbS polymerisation during the  transit of erythrocytes through post‐capillary venules is necessary for SCA disease remediation [56]."

Legend of Figure S3. Ddata change to Data

Legend Figure S8 the Font size is smaller than in other Figures. Change to the same Font size.
